# Hierarchical Gated Recurrent Unit with Semantic Attention for Event Prediction

## Zichun Su * and Jialin Jiang

School of Computer Engineering and Science, Shanghai University, Shanghai 200444, China;
nirvanawings@shu.edu.cn

* Correspondence: zc_su@shu.edu.cn; Tel.: +86-187-0197-2809

**Abstract:** Event prediction plays an important role in financial risk assessment and disaster warning, which can help government decision-making and economic investment. Previous works are mainly based on time series for event prediction such as statistical language model and recurrent neural network, while ignoring the impact of prior knowledge on event prediction. This makes the direction of event prediction often biased or wrong. In this paper, we propose a hierarchical event prediction model based on time series and prior knowledge. To ensure the accuracy of the event prediction, the model obtains the time-based event information and prior knowledge of events by Gated Recurrent Unit and Associated Link Network respectively. The semantic selective attention mechanism is used to fuse the time-based event information and prior knowledge, and finally generate predicted events. Experimental results on Chinese News datasets demonstrate that our model significantly outperforms the state-of-the-art methods, and increases the accuracy by 2.8%.

**Keywords:** event prediction; semantic selective attention; associated link network

## 1. Introduction

As Mark Twain said, the past does not repeat itself, but it rhymes. We may be able to learn the evolution of events from past events. Taking the news reports of *"2007 Peru earthquake"* and *"2010 Chile earthquake"* as an example (see Figure 1), the same event background and evolution path exist in news reports from different periods (events are all evolved along with the background of *"earthquake"*, *"tsunami warning"*, *"personal casualty"*, and *"disaster assistance"*). This demonstrates that similar events have similar backgrounds, and historical events may provide a wealth of prior knowledge for event prediction. Event prediction is very important for government decisions [1–3] and financial investment [4,5]. Governments can avoid casualties and property damage based on disaster warnings. At the same time, they also can predict social trends and understand public opinion. Financial institutions can estimate investment risks to avoid potential economic losses.

Because of the high application value of event prediction, research interest in this field is increasing. Some research uses traditional machine learning methods for event prediction. These methods apply textual characteristics of events, statistical language models, etc. for event prediction. For example, Yang et al. [6] used the timestamp, text similarity, and temporal proximity to infer the event evolutional relationship. Manshadi et al. [7] proposed a method for event prediction based on an n-gram language model. In recent years, many research attempts to use neural network in event prediction tasks. Most methods apply Recurrent Neural Network (RNN), a neural network that processes sequence data, for event prediction. Pichotta et al. [8] described a Long Short-Term Memory (LSTM) model for statistical script learning. Wang et al. [9] used LSTM hidden states as features for event pair modeling and utilized a dynamic memory network to infer subsequent events.

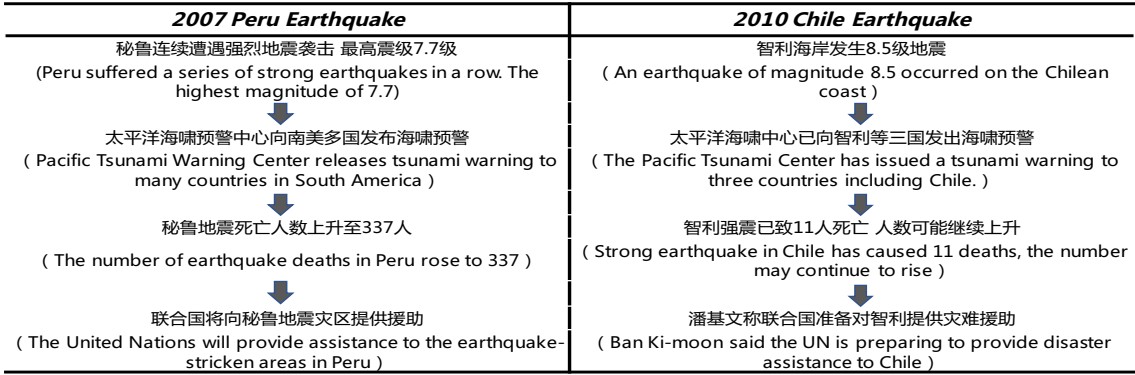

| *2007 Peru Earthquake* | *2010 Chile Earthquake* |
|---|---|
| 秘鲁连续遭遇强烈地震袭击 最高震级7.7级<br>(Peru suffered a series of strong earthquakes in a row. The highest magnitude of 7.7) | 智利海岸发生8.5级地震<br>( An earthquake of magnitude 8.5 occurred on the Chilean coast ) |
| 太平洋海啸预警中心向南美多国发布海啸预警<br>( Pacific Tsunami Warning Center releases tsunami warning to many countries in South America ) | 太平洋海啸中心已向智利等三国发出海啸预警<br>( The Pacific Tsunami Center has issued a tsunami warning to three countries including Chile. ) |
| 秘鲁地震死亡人数上升至337人<br>( The number of earthquake deaths in Peru rose to 337 ) | 智利强震已致11人死亡 人数可能继续上升<br>( Strong earthquake in Chile has caused 11 deaths, the number may continue to rise ) |
| 联合国将向秘鲁地震灾区提供援助<br>( The United Nations will provide assistance to the earthquake-stricken areas in Peru ) | 潘基文称联合国准备对智利提供灾难援助<br>( Ban Ki-moon said the UN is preparing to provide disaster assistance to Chile ) |

**Figure 1.** News reports on the "2007 Peru Earthquake" and the "2010 Chile Earthquake".

Although the above methods have achieved good success, there are still some shortcomings of prediction. Firstly, current methods based on event pairs can only judge the evolution relationship between events that already exist in the training data, while it is difficult to represent the evolution direction of events. Secondly, time series-based event prediction may ignore the influence of earlier events, which will cause some bias in the direction of event evolution. Finally, the event description has few words, which cannot contain the rich semantics of events. How can we supplement the event description?

To overcome these problems, we propose a novel model, called Hierarchical Semantic Gated Recurrent Unit (HS-GRU), for event prediction. HS-GRU extracts time-based event information and prior knowledge of current events separately, and fuses them by the semantic selective attention mechanism to predict the next possible event. The input of HS-GRU is an event link which is the sequence of related events in a short period. Of these, the current event link for prediction provides time-based event information, and historical event links provide prior knowledge for the current event prediction. The output of model is a textual description of the next event. HS-GRU includes three layers. The first layer encodes time-based event information and prior knowledge of the current event link. Among them, the model adopts Gated Recurrent Unit (GRU) [10] to collect time-based event information in the current event link. We use Associated Link Network (ALN) [11] to capture prior knowledge of the current event link. ALN extracts the semantic background related to the current event link from historical event links as the prior knowledge of event prediction. The second layer fuses time-based event information and prior knowledge of the current event link. The model applies the semantic selective attention mechanism to calculate the impact score of events in the current event link under the prior knowledge. The score is then used to weight the time-based event information to get the current event link embedding, rather than just relying on timing. The last layer takes the event link embedding as input and generates the next possible event by GRU decoder.

The contributions of this paper are as follows:

1. We construct a novel HS-GRU model that predicts events based on time-based event information and prior knowledge of current event link.
2. We propose a semantic selective attention mechanism to combine temporal information with prior knowledge. It calculates the influence score of current events on the next event under the prior knowledge and fuses the event information by the score.
3. Experimental results on Chinese News datasets demonstrate that our model is superior to several state-of-the-art methods on event prediction tasks.

The code and dataset are publicly available at https://github.com/bubble528/HS-GRU. The rest of this paper is structured as follows. We start by describing the previous related work. Section 3 presents the proposed method of HS-GRU. The experiment results and analysis are shown in Section 4. Finally, we summarize our work in Section 5.

## 2. Related Work

Event prediction is a challenging task. At present, event prediction methods are mainly divided into methods based on traditional machine learning and deep learning.

### 2.1. Traditional-Machine-Learning-Based Methods

Some current research used textual characteristics of events to prediction. Yang et al. [6] utilized event timestamps, event content similarity, temporal proximity, and document distribution proximity to model event evolutional relationships between events and constructed an event evolution graph. To accommodate the rapid growth of social media, Weiler et al. [12] proposed a simple sliding window model, which uses the offset of inverse document frequency (IDF) to capture the trend of keywords and track the evolution of events. Lu et al. [13] proposed a method to automatically discover the evolution of microblog events based on multiple similarity measures such as contents, locations, and participants. It establishes a 5-tuple event description model for events detected in microblog and analyzes the relationship between them. Zhou et al. [14] used Term Frequency $\times$ Inverse Document Frequency (TF $\times$ IEF) and Time Distance Cost factor (TDC) to model the event evolutional relationship. Liu et al. [15] proposed a Knowledge-based Event Mining (KEM) model. It merges the same entities and similar phrases by the knowledge base and incremental word2vec model. It uses a 7-tuple event description model to display the events comprehensively and analyze the relationship between events and incrementally generate an evolutional link for each event. Finally, the model generates an evolutional link for each event. There are also some probability-based methods. Manshadi et al. [7] proposed a method for event prediction based on the n-gram language model. Jans et al. [16] used bigram to model the temporal order of event pairs and formed event links for event prediction. Radinsky et al. [17] predicted events by causality between events. The method extracts the generalized causality of two events in the form of "x cause y" and applies the template to a current news event to predict the next event. Li et al. [18] proposed Event Evolution Graph (EEG), which constructs by identifying the sequential relationship between events and the direction of each sequential relationship, to reveal evolutional patterns and development logic between events.

### 2.2. Deep-Learning-Based Methods

Nowadays, many researchers applied deep learning to event prediction tasks. Granroth-Wilding et al. [19] described a compositional neural network. It simultaneously learns the function of event representation and the coherence function that predicts the strength of association between two events, which is used to predict whether the event is likely to be the next event. Pichotta and Mooney [8] used LSTM for statistical script learning to predict future events, where the script is a sequence of prototype events represented by multiple predicates. Wang et al. [9] utilized LSTM to model event pairs by event sequence information and relationship of event pairs. Hidden states of LSTM are used to calculate the correlation score of event pair to infer subsequent events. Hu et al. [20] built an event sequence prediction model based on a hierarchical LSTM combined with a topic model. Li et al. [21] used GNN to encode event evolution graphs extracted from a large number of news corpora and selected the most reasonable future events by comparing the similarities between context events and candidate events.

## 3. Our Proposal

Most of current methods are based on RNN for event prediction, which can only extract the sequence information of events. Here, we propose a hierarchical event prediction model based on time series and prior knowledge. The model extracts the time-based event information in event descriptions by GRU and provides the prior knowledge of current events link by ALN. The semantic selective attention mechanism is used to fuse time-based event information and prior knowledge and guide the event prediction. Among them, ALN extracts the semantic background of current event

link by associated relation among words in event links, which is used as the prior knowledge of event prediction.

Our model takes event links as input for prediction. An event link consists of a sequence of related events that occur over a short period. The current event link which is being used for prediction provides time-based event information. All historical event links provide prior knowledge. At first, we define the event and the event link included in the model. We consider that an event $E = \{w_1, \cdots, w_N\}$ is a sequence of words, where $N$ is the length of event $E$. An event link $L = \{E_1, \cdots, E_M\}$ is a sequence of $M$ events composed in temporal order. Note that a special token $\langle \text{end} \rangle$ is added at the end of each event.

### 3.1. Gated Recurrent Unit

Gated Recurrent Unit (GRU) [10] is used to encode time-based event information. GRU is a variant of Recurrent Neural Network (RNN) [22], which is a type of neural network for processing sequence data. The original RNN cannot capture long sequence information because of long-term dependencies [23]. GRU solves the problem of long-term dependencies with a gate function and is now widely used in real-world applications.

The event we defined is represented in the form of a sequence of words. There is naturally a time series between words in the expression of an event. The description of predicted events generated by our model also has the word order relationship. Therefore, we construct a hierarchical architecture model for event prediction based on GRU. It is the basic unit of calculation in the HS-GRU.

### 3.2. Association Link Network

Association Link Network (ALN) [11] can provide prior knowledge for event prediction. The input of our model is the description of events. However, the current event description contains fewer words, which makes it difficult to express the event semantics completely. Therefore, we add prior knowledge to complement the event semantic description. Historical events contain rich semantics, and current events are often repetitions of historical events, which contain similar semantics. Thus, we apply the semantic background which related to the current event as the prior knowledge for event prediction. It can supplement the missing information of the event description itself. We apply historical event links to construct a semantic network, that is, the Association Link Network (ALN) [11]. ALN extracts the semantic background related to the current event link, which provides prior knowledge for event prediction.

ALN is a method to organize the semantics that is loosely distributed and sparsely associated in text. It aims to organize information by establishing the relationship between resources. Cognitive science theory considers that the association can be more comprehensive for users to understand resources. The motivation of using ALN is to organize the associated semantic information of event links and guide the evolution of events.

We believe that topics of related event links are consistent, thus words that often appear in the identical top will naturally be associated and aggregated. The associated relation among words in events will reflect the semantic relationship of events. Based on the above ideas, we organize the event semantics by the co-occurrence relationship between words in all historical event links and construct an ALN to provide prior knowledge for event prediction. The basic components of the ALN are semantic nodes and associated links connecting semantic nodes. The semantic node refers to the word in an event link, and the associated link refers to the co-occurrence relationship between words. The formula for calculating the associated link weight $A^{w_i - w_j}$ between the word $w_i$ and the word $w_j$ is:

$$A^{w_i - w_j} = \frac{\text{Co}\left(w_i, w_j\right)}{\sqrt{DF\left(w_i\right) \cdot DF\left(w_j\right)}} \tag{1}$$

where $\mathrm{Co}\left(w_i, w_j\right)$ is the frequency of $w_i$ and $w_j$ co-occurrence in one event link and $DF\left(w_i\right)$ represents the frequency of word $w_i$ in event links. The associated link weight $A^{w_i - w_j} \in [0, 1]$ indicates semantic association strength between words.

### 3.3. Hierarchical Semantic Attention

Using GRU and ALN, we can obtain the time-based event information and prior knowledge of events, which is the basis of our event prediction. Combining the time series and the prior knowledge, we propose HS-GRU for event prediction, as illustrated in Figure 2. The model extracts the time-based event information by GRU and obtains the semantics of current event link by ALN. The semantics provides prior knowledge for prediction. The model finally fuses the time-based event information and semantic background using the semantic selective attention mechanism to predict the next event.

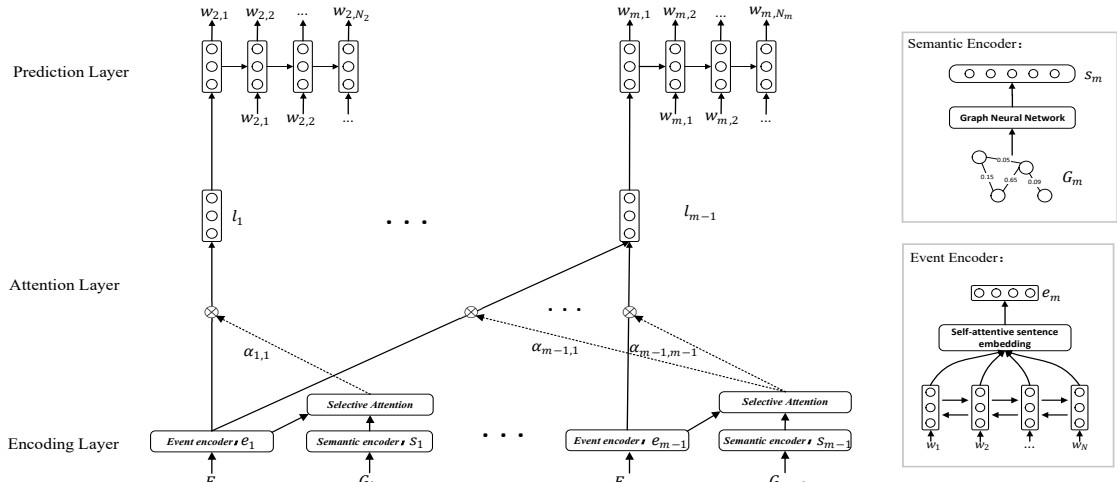

**Figure 2.** Illustration of our proposed HS-GRU model for event prediction. The encoding layer obtains the time-based event information and prior knowledge of current event link. The attention layer fuses time-based event information and prior knowledge by semantic selective attention mechanism. The prediction layer is used to generate the next event text description.

The model consists of three layers: encoding layer, attention layer, and prediction layer. The encoding layer encodes time-based event information and the semantic background of the current event link. The attention layer applies semantic selective attention mechanism to calculate the influence score of current events under the semantic background and generate the current event link embedding. Finally, the prediction layer generates the text description of next event.

#### 3.3.1. Encoding Layer

The encoding layer mainly uses GRU and ALN to calculate time-based event information and prior knowledge of the current event link, respectively. It is composed of two parts: event encoder and semantic encoder.

Event encoder maps the word sequence of events in the input event link to event vectors by GRU, without using manually extracted features. To obtain a more comprehensive event embedding, we use bidirectional GRU to obtain event embeddings through the structured self-attentive mechanism [24]. For an event $E = \{w_1, \cdots, w_N\}$, we encode an event as follows:

$$\overrightarrow{h_n^e} = \overrightarrow{GRU}\left(w_n, \overrightarrow{h_{n-1}^e}\right); \overleftarrow{h_n^e} = \overleftarrow{GRU}\left(w_n, \overleftarrow{h_{n+1}^e}\right) \tag{2}$$

$$\boldsymbol{h_n^e} = \left[\overrightarrow{h_n^e}, \overleftarrow{h_n^e}\right], \tag{3}$$

$$H^e = (\boldsymbol{h_1^e}, \boldsymbol{h_2^e}, \cdots, \boldsymbol{h_N^e}) \tag{4}$$

$$A = \text{softmax} \left( W_{s2} \tanh \left( W_{s1} H^{e^T} \right) \right) \tag{5}$$

$$e = \tanh \left( w_f \left( A H^e \right) + b_f \right) \tag{6}$$

where $\overrightarrow{GRU}$ is a function of encoding word sequences with unidirectional GRU, $\overrightarrow{h_n^e} \in \mathbb{R}^{d_e}$ denotes the unidirectional hidden state, and $\overrightarrow{h_{m,0}^e} = 0$, $\boldsymbol{h_n^e}$ is the hidden state at the nth position connected by $\overrightarrow{h_n^e}$, $\overleftarrow{h_n^e}$, $A$ is a weight matrix of the structured self-attention mechanism, and $W_{s1}$ and $W_{s2}$ are parameter matrices. $tanh()$ is an activation function. $softmax()$ ensures that the sum of weights equals 1. Finally, the embedding $e$ of the event $E$ is obtained from the product of the weight matrix and the hidden vector by a fully connected layer whose parameters are $w_f$ and $b_f$.

The semantic encoder mainly encodes the prior knowledge of the current event link, which is the semantic background. We extract the semantic background in event links by ALN, which provides prior knowledge for prediction. We build an ALN with historical event links, which are all event links in the training dataset. The ALN contains all semantics of historical event links, thus the network is large. However, the prior knowledge associated with the current event link is only a subgraph in ALN. Therefore, for the current event link, we extract a sub-ALN related to the current event link in ALN as the prior knowledge of event prediction. The sub-ALN contains words in the current event link, first neighbors of these words in ALN, and the edges connected to these words. However, sub-ALN is a network structure based on word association, which cannot be directly used for calculation, thus we need to encode the network as a high-dimensional vector. To encode a network, we employ Graph Neural Networks (GNN), a graph encoding technique, to obtain an embedding of sub-ALN associated with the current event link. Scarselli et al. [25] first proposed the basic graph neural network (GNN), extending neural network for computing graph-structured data. Li et al. [26] introduced gated units on GNN and proposed gated GNN, which is the structure we used in our model. GNN encodes the current node by adjacent nodes, so that it automatically extracts features of graphs with rich node connections. GNN takes account of the characteristics of graphs, which is also suitable for sub-ALN. We present the learning process of node vectors in a sub-ALN. In a sub-ALN $G$, the encoding process of node $w_i$ is as follows:

$$a_i^t = C_{i:} \left[ v_1^{t-1}, \ldots, v_n^{t-1} \right]^\top H + b \tag{7}$$

$$z_i^t = \sigma \left( W_z a_i^t + U_z v_i^{t-1} \right), \tag{8}$$

$$r_i^t = \sigma \left( W_r a_i^t + U_r v_i^{t-1} \right), \tag{9}$$

$$\tilde{v}_i^t = \tanh \left( W_o a_i^t + U_o \left( r_i^t \odot v_i^{t-1} \right) \right), \tag{10}$$

$$v_i^t = \left( 1 - z_i^t \right) \odot v_i^{t-1} + z_i^t \odot \tilde{v}_i^t \tag{11}$$

where $H \in \mathbb{R}^{d \times d}$ is a weight matrix; $\left[ v_1^{t-1}, \ldots, v_n^{t-1} \right]$ is a vector list of nodes in sub-ALN; $\mathbf{z}_i^t$ and $\mathbf{r}_i^t$ are the update gate and the reset gate, respectively; $\sigma(\cdot)$ denotes the logistic sigmoid function; $\odot$ is the element-wise multiplication; and $v_i^t$ stands for the hidden state vector of node at time step $t$. The adjacency matrix $C \in \mathbb{R}^{n \times n}$ determines how nodes in sub-ALN are associated with other nodes.

The sparse structure and parameter binding of adjacency matrix $C$ are shown in Figure 3. The sparse structure of the matrix $C$ is determined by edges in the sub-ALN, and parameters of the matrix are judged by the weight of edges. The value of corresponding position in adjacency matrix $C$ is the value of associated weight of two points. Please note that, depending on event links, the number of nodes included in sub-ALN is different.

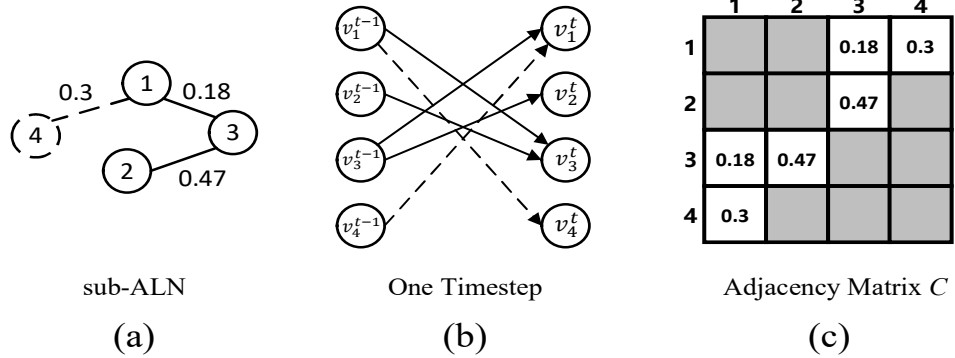

**Figure 3.** (**a**) Example sub-ALN. Solid lines are words in the event link, and dotted lines are first-order neighbors. (**b**) Unrolled one timestep. (**c**) Parameter binding of adjacency matrix. The parameter is the weight of corresponding edge in sub-ALN.

For each sub-ALN $G$, GNN processes nodes synchronously. After all node vectors in the network are updated until convergence, a semantic embedding $s$ of a sub-ALN $G$ is obtained:

$$s = \frac{1}{N_G} \sum_{i=1}^{N_G} v_i'$$

(12)

where $v_i'$ is the final vector of a node in $G$ and $N_G$ is the number of nodes in $G$.

### 3.3.2. Attention Layer

For current event link $L = \{E_1, E_2, \cdots, E_M\}$, the attention layer mainly fuses the time-based event information and the semantic background of current event link by the semantic selective attention mechanism, and finally generates the current event link embedding. In the past, the RNN-based approach only focused on the time series of event links. Less attention has been paid to the impact of event semantic background on the event evolution direction. We consider that the influence of different semantics on the evolution direction of events is discrepant. Therefore, we propose a semantic selective attention mechanism, which is a method for calculating attention weight by the semantic background. Under the semantic background of current event link, we apply the semantic selective attention mechanism to calculate impact weights of current events on prediction. This can pay more attention to events that have a significant impact on the development direction of event links. The attention layer calculates the influence score of events in the current event link by the selective attention mechanism, and weights event embeddings to get event link embedding $l$. Influence score $\alpha_i$ of event $E_i$ is defined as follows:

$$\alpha_i = \frac{\exp(u_i)}{\sum_m \exp(u_m)}$$

(13)

where $m$ stands for the number of events in event link $L$. $u_i$ is a scoring function, which scores the degree of matching between event embedding $e_i$ and event link semantic embedding $s$. We chose the bilinear form as follows:

$$u_i = e_i W s$$

(14)

where $W$ is a weight matrix. Finally, the influence score is used to evaluate the effect of current events on event prediction under semantics $s$ and compute the event link embedding $l$. The process is as follows:

$$l = \Sigma_{i=1}^m \alpha_i e_i$$

(15)

where *l* is the event link embedding under semantics *s*. The event link finally is embedded by weighting event embeddings with influence score $\alpha$.

### 3.3.3. Prediction Layer

The last layer is responsible for predicting the next event $E_m$ based on previous events $E_{1:m-1}$, i.e., the probability $P\left(E_m|E_{1:m-1}\right)$:

$$P\left(E_m|E_{1:m-1}\right) = \prod_{n=1}^{N_m} P\left(w_{m,n}|w_{m,1:n-1}, E_{1:m-1}\right) \tag{16}$$

After processing by two layers, we get an event link embedding *l*. A GRU decoder is applied to generate the next event $E_m$ verbatim. The decoder, which is similar to the encoder, is as follows:

$$h_{m,n}^d = GRU\left(h_{m,n-1}^d, w_{m,n-1}\right), n = 1, \cdots, N_m \tag{17}$$

$$P\left(w_{m,n} = v|w_{m,1:n-1}, e_{1:m-1}\right) = \frac{\exp\left(\boldsymbol{o}_v^T \omega\left(h_{m,n-1}^d, w_{m,1:n-1}\right)\right)}{\exp\left(\sum_k \boldsymbol{o}_k^T \omega\left(h_{m,n-1}^d, w_{m,1:n-1}\right)\right)} \tag{18}$$

where *GRU* is a function of decoding a sequence of words describing a possible next event, $h_{m,0}^d = l_{m-1}$, where $l_{m-1}$ is the embedding of event link $L = \{E_1, \cdots, E_{M-1}\}$. Following Sordoni et al. [27], we use a softmax layer to compute a hidden state $h_{m,n}^d$ to generate the next possible word, as shown in Equation (18), where $O_v \in \mathbb{R}^{D \times V}$ is the output embedding of word *v*. $\omega(\cdot)$ is a linear transformation function to generate the word. This formula has proven to be beneficial for language model tasks [28].

### 3.4. Learning

Model parameters are primarily learned by maximizing the log-likelihood of event links, defined by Equation (19):

$$\begin{aligned}
\mathcal{L}(L) &= \sum_{m=1}^{M} \log P\left(E_m|E_{1:m-1}\right) \\
&= \sum_{m=1}^{M} \sum_{n=1}^{N_m} P\left(w_{m,n}|w_{m,1:n-1}, E_{1:m-1}\right)
\end{aligned} \tag{19}$$

where *M* is the number of events in an event link and $N_m$ is the number of words in event $E_m$. Finally, we use the Back-Propagation Through Time (BPTT) algorithm to train the proposed HS-GRU model.

## 4. Experiments

### 4.1. Dataset

Since important breaking events are usually first reported in the news, we crawled our dataset from Sina News (http://news.sina.com.cn/zt/), one of the largest Chinese new websites. The news is classified into several categories: China news, international news, social news, military news, financial news, etc. News in each category are organized according to different subjects (e.g., "*2013 German Bundestag election*" and "*Egyptian hot air balloon explosion*"), and each subject is composed of an article sequence in temporal order. The website contains 9201 news subjects from 1998 to 2015, and each news subject contains an average of 60 events. We only use news headlines because they are the core of news. To verify the generalization ability of the model, we constructed international news and Chinese news datasets for experiments. Because the number of China news in the website is more than international news, the two datasets have different sizes.

Events in each subject constitute an event link, but too many events in one subject will make the event link too long, which will reduce the accuracy of prediction. Prediction based on long event

sequences is of minor significance. Following Hu et al. [20], the dependency between more than five events is minimal, thus a sliding window of size 5 was used to divide events under a subject into event links. We further processed events in event links. We utilized HanLP (http://hanlp.com/) to segment events, and then removed stop words in word segmentation results according to the stop word lists of Harbin Institute of Technology, Sichuan University Machine Intelligence Lab and Baidu (https://github.com/baipengyan/Chinese-StopWords). After that, to reduce data sparsity, the low-frequency words were deleted. Finally, we randomly divided each set of data into training set, validation set, and test set according to the percentages of 80%, 10%, and 10%. The data details are shown in Table 1.

**Table 1.** Statistics of datasets.

|  | International News | China News |
|---|---|---|
| Events | 120,879 | 259,327 |
| Event links | 114,785 | 237,480 |
| Words | 2521 | 3538 |
| Training | 91,824 | 189,984 |
| Validation | 11,480 | 23,748 |
| Test | 11,481 | 23,748 |

### 4.2. Model Training

We set parameters as follows. All parameters were first initialized using a normal distribution with the standard deviation N (0, 0.01). The batch size was 32, the learning rate was $10^{-4}$, and the dimension of word embedding was 512. For the model, the dimension of event encoder GRU and event prediction GRU was 512, and the dimension of GNN encoder was 512. The number of key points in the attention mechanism of the event encoder was 5. The weight of the edge in ALN was (0.1, 1).

We used Ttensorflow 1.8 to build our prediction model and applied GTX 1080Ti for model training. The training details are shown in Table 2.

**Table 2.** Training detail.

|  | Batches | Training |
|---|---|---|
| International News | 837,800 | 71 h 43 min |
| China News | 894,300 | 75 h 23 min |

### 4.3. Evaluation

There is no well-established method for event prediction, and human-based evaluation is expensive. Referring to Serban et al. [29], error rate, the probability of word misclassification, was used to evaluate our model. It is defined as the proportion of the number of error words in the prediction result to the total number of words in the sentence.

$$ErrorRate = \frac{1}{N_w} \sum_{m=1}^{M^l} \sum_{i=1}^{N_m^e} I\left(w_{m,i}^{\prime e} \neq w_{m,i}^e\right) \tag{20}$$

where $I(\cdot)$ is an indicator function; when the result of a word prediction is wrong, i.e., $w_{m,i}^e \neq w_{m,i}^e$, the function is equal to 1. $w_{m,i}^{\prime e}$ indicate the $i$-th word predicted in the event $e_m$. $w_{m,i}^e$ stands for the real $i$th word in $e_m$. $N_m^e$ is the number of words of $e_m$. $M^l$ is the number of events in the event link $L$. $N_w$ denotes the total number of words in the event link $L$. Notice that, when predicting a word, only when the word and the position are both correct, the result is correct. A low error rate for the model indicates that the prediction model is more accurate.

Our model was compared to following baseline methods:

1. **N-Gram** is a language model which assumes that the occurrence of the $N$th word is only related to the previous $N - 1$ words. It uses probabilistic methods to reveal the statistical laws inherent in language units. Here, we used two n-gram with different smoothing techniques, Backoff N-Gram and Modified Kneser-Ney [30], to compare with our model.
2. **GRU** [10] is a variant of RNN that has a simpler structure for the long-term dependencies of RNN labeling. It can do well with sequence problems.
3. **HRED** [27] is a hierarchical encoder–decoder suggestion model that encodes the context information of the query layer and the session layer, respectively, and finally provides context-aware query suggestions for users. It is similar to the task of our model.

Here, the parameter n of n-gram is 3. For neural network models, their parameters are the same. Experimental results are given in Table 3.

**Table 3.** Error rate on datasets.

| | Error Rate | |
| --- | --- | --- |
| | **International News** | **China News** |
| Backoff N-Gram [Goodman, 2001] | 96.71% | 95.94% |
| Modified Kneser-Ney [Goodman, 2001] | 93.52% | 94.75% |
| GRU [Cho et al. 2014] | 54.44% | 66.47% |
| HRED [Sordoni et al. 2015] | 51.30% | 64.57% |
| HS-GRU (ours) | 48.48% | 62.33% |

In general, because international news has fewer words than China news, error rates of international news are lower than China news. Our results demonstrate that the accuracy of language models is extremely low. Error rates of two n-gram models are much higher than deep learning methods. It demonstrates that the traditional language model, which relies only on the probability of occurrence between n-words, has difficulty meeting the needs of the task. Compared with the language model, deep learning methods can better adapt to the prediction task, and their accuracy are greatly improved. For the deep learning model, the hierarchical RNN model, HRED, is more accurate than the simple single-layer GRU model in our dataset. This manifests that, in the event link-based prediction task, the hierarchical model that processes the event information and event link information hierarchically can extract the sequence information of the event better than the single-layer model. However, both models only use RNN to obtain the time-based information of events for prediction. For our model, compared with the HRED, error rates decrease by 2.82% and 2.24%. It shows that adding the prior knowledge to time-based event information can help to improve the accuracy of event prediction task. Compared with previous models, the prediction accuracy of HS-GRU has been improved. It indicates that adding prior knowledge can modify the prediction result which used only time-based information. Prior knowledge of current event links plays an important guiding role in event prediction.

In datasets, lengths of event descriptions are different. To verify the predictive performance of our model for different length events, we divided the event link into long and short event links to compare the performance of HS-GRU. We took event links with length in the top 30% of datasets as short event links, and the rest as long event links. We compared our model with the basic model HRED. The results are shown in Table 4. Experimental results demonstrate that, for both international news and China news, error rates of short event links are higher than those of long event links. This is because short event links contain incomplete descriptions of events, which can easily cause deviations in event prediction. For the same dataset, compared with the basic model, the error rate optimization percentage ($\Delta\%$) of long event links is higher than short event links. This manifests that our model is more capable

of extracting long-range information from event links. Based only on time-based information, the model may forget past events. Prior knowledge can provide more guidance information for event prediction. Our model uses prior knowledge provided by historical events to determine the importance of current events to predictions. It improves the accuracy of predictions.

**Table 4.** Error rate for events link with different length.

|  | International News | | China News | |
| --- | --- | --- | --- | --- |
|  | Long Event Link | Short Event Link | Long Event Link | Short Event Link |
| HRED | 45.63% | 68.32% | 59.51% | 73.23% |
| HS-GRU | 42.50% | 66.41% | 57.43% | 71.42% |
| Δ% | 3.13% | 1.91% | 2.48% | 1.8% |

We used HS-GRU to predict the possible next event. For current event link $L = \{E_1, E_2, \cdots, E_m\}$, the model generates $E_m$ given $E_{1:m-1}$. Taking the news event links of the Indonesian tsunami as an example, the output of different models is shown in Figure 4. The results manifest that our model is the closest to the real result. Among predictions of each model, although predicted results of GRU and HRED are related to *"earthquake"* and *"Chinese"*, they do not predict the information about *"kill"*. In the event link on the left of Figure 4, *"kill"* was mentioned in the first event. However, in GRU and HRED, they pay more attention to information such as *"earthquake"* and *"tsunamis"*. Our model predicts "kills", which further shows that adding prior knowledge can make event prediction more accurate.

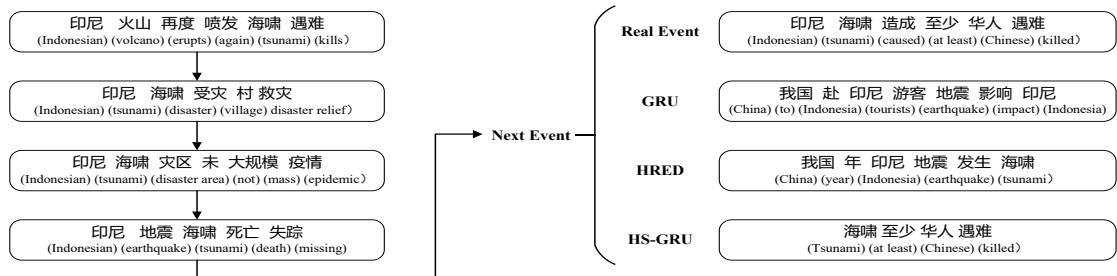

**Figure 4.** An example of model outputs. Previous four events in the event link are shown on the left. The real last event and the prediction results for different model are shown on the right.

*4.4. Event Selection*

Generative model-based event prediction is a difficult task, which requires predicting the correct sequence of all words in the text. In real-world applications, the low accuracy of prediction results makes it difficult to apply models to specific prediction tasks. Because of the diversity of languages, sentences with a high prediction error rate may express correct semantics. At the same time, sentences generated by the model are often poorly readable. Therefore, we applied the model to the event selection task. Given a series of candidate real events, the model calculates the similarity between the candidate event and the predicted result to find the most likely next event. An example of this task is shown in Figure 5. Here, the similarity between the candidate event and the predicted result was evaluated by the Jaccard index, defined as follows:

$$J\left(E_i, E_j\right) = \frac{\left|W_{E_i} \cap W_{E_j}\right|}{\left|W_{E_i} \cup W_{E_j}\right|} \tag{21}$$

where $W_{E_i}$ is the set of words in event $E_i$. Based on the above ideas, our model was applied to the next event selection task that aims to estimate whether the model can correctly hit next events.

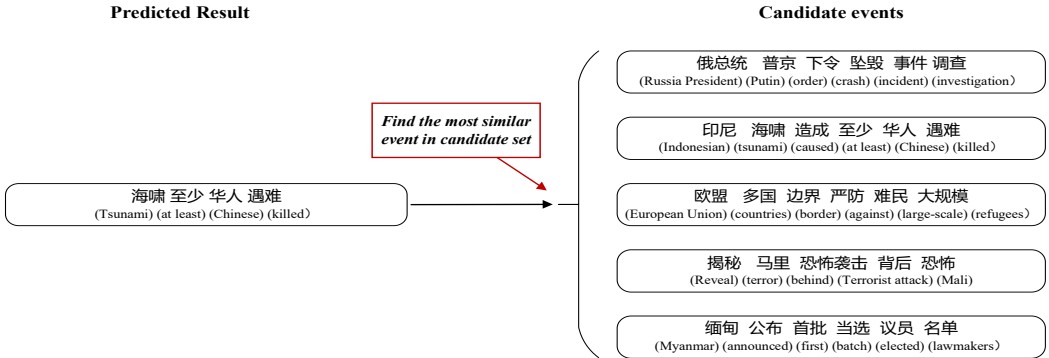

**Figure 5.** Event selection example. It verifies whether the model can find the correct event in the candidate set by the prediction result.

Referring to the method of Ghosh et al. [31], the candidate event set was built in the test set. The test set was randomly divided into subsets, each of which contained 50 event links. For each subset, the last event of each event link in the subset was selected as a candidate event. Finally, each event link in subsets had a candidate event set of size 50. We used the hit rate to evaluate different models. We compared the *hit rate* of current result in candidate events with similarity in the Top 1, Top 5, and Top 10. Table 5 shows experimental results.

**Table 5.** The result of event selection task on two datasets.

|  | International News | | | China News | | |
|---|---|---|---|---|---|---|
|  | Top 1 | Top 5 | Top 10 | Top 1 | Top 5 | Top 10 |
| Random | 2.00% | 10.00% | 20.00% | 2.00% | 10.00% | 20.00% |
| GRU | 56.00% | 74.36% | 78.00% | 49.27% | 69.27% | 71.64% |
| HRED | 59.64% | 74.73% | 79.64% | 53.82% | 71.45% | 73.45% |
| **HS-GRU** | 69.45% | 82.73% | 86.36% | 56.36% | 75.82% | 78.91% |

Compared to the GRU and HRED models, the hit rate of our model is also improved in the two datasets. It further proves the importance of the combination of time-based information and prior knowledge for event prediction, which also improves the application value of the model.

## 5. Conclusions

Event prediction has important value in government decision-making and financial investment. Previous methods are mainly based on the time series of events for prediction, without considering the impact of prior knowledge on event prediction. In this paper, we construct a novel model, called Hierarchical Semantic Gated Recurrent Unit (HS-GRU), which is a hierarchical event prediction model based on time series and prior knowledge for the task of events prediction. HS-GRU adopts the semantic selective attention mechanism to fuse time-based event information and prior knowledge for event prediction. It does not need hand-crafted event features, which can be applied in many domains. Our model consists of three layers. The encoding layer encodes the time-based event information and the prior knowledge of the current event link by Gated Recurrent Unit (GRU) and Association Link Network (ALN), respectively. The attention layer adopts the semantic selective attention mechanism to fuse time-based event information and prior knowledge and calculates the current event link embedding, instead of relying on time series. The prediction layer takes the event link embedding as an input and generates the next event with GRU decoder. Experiments demonstrated that the accuracy of our model is significantly better than the state-of-the-art methods. Because the model does not have domain features, it has important value in multiple applications such as emergency warning,

traffic forecast, stock prediction, and disease spreading. In the future, we will consider abandoning the traditional Recurrent Neural Network (RNN) model for processing sequence structures and discuss the method for predicting events using semantic networks which encoded by the Graph Neural Network.

**Author Contributions:** methodology, Z.S.; validation, Z.S.; writing—original draft preparation, Z.S; resources, J.J.; writing—review and editing, Z.S. and J.J. All authors have read and agreed to the published version of the manuscript.

**Funding:** This research received no external funding.

**Conflicts of Interest:** The authors declare no conflict of interest.

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
