# Peer review of "Hierarchical Gated Recurrent Unit with Semantic Attention for Event Prediction"

_futureinternet, doi:10.3390/fi12020039_

Round 1
Reviewer 1 Report
I find this paper very interesting and worthy publication even though, like Authors admit in the Results section, it is just another small step in the field.
So in my case this is a Minor Revision only
Please add section about hardware/software environment. Is this tool availbale as some package / Github project maybe? What was the software used for all your calculations? What was the computational workload required to achieve these results? If possible please publish curated dataset used in your development. This would allow others to compare directly to your relsults and increase visibility of your paper Have you considered ensembling your system? I.e. work with collective answers of many models for the final prediction. Maybe it could improve results? Please comment on this In line 296 probably should be "... It can do well with sequence problems."Author Response
Point 1: Please add section about hardware/software environment. Is this tool available as some package / Github project maybe? What was the software used for all your calculations? What was the computational workload required to achieve these results? 

Response 1: We use tensorflow1.8 to build our prediction model and apply GTX 1080Ti to train the model. For international news, the model trains 837,800 batches for a total of 72 hours. For domestic news, because the amount of data is greater than international news, a total of 894,300 batches are trained for 75 hours. In line 281 of the pdf file, we added the training environment and explain it in detail in Table 2. The source code of the model has not been uploaded to GitHub before. Now, the source code and dataset have been uploaded to GitHub. URL link added in line 66 on page 2 of PDF file.
Point 2: If possible please publish curated dataset used in your development. This would allow others to compare directly to your results and increase visibility of your paper.
Response 2: We have published the source code and dataset on GitHub. The link is shown in line 66 on page 2 of PDF file.
Point 3: Have you considered ensembling your system? I.e. work with collective answers of many models for the final prediction. Maybe it could improve results?
Response 3: The main task of our model is to predict the sequence of future events. At present, the more effective method of sequence prediction is RNN. Our model adds prior knowledge to the time-based information extracted by RNN, which supplements the semantic information while retaining the characteristics of the RNN method. This combines features of different methods. In a way, it is also a combination of multiple models. Therefore, we did not use assembly systems for final prediction.
Point 4: Please comment on this In line 296 probably should be "... It can do well with sequence problems."
Response 4: In line 300 of the PDF file, we modified this sentence to " It can do well with sequence problems. "

Reviewer 2 Report
This paper addresses the problem of event prediction, which is of interests to the community. It improve existing approaches from two main aspects: 1. it proposes to build a hierarchical model which take prior knowledge into consideration, rather than a simple time based autoregressive model; 2. it uses attention mechanism to obtain attention between event and prior, which is a novel and useful method to build cross relationship thus enhance prediction results.
While this is an improved RNN based approach, the presentation is clear and the experiment details are well provided. Some part could be further improved as follows:
For evaluation, some other metric such as dynamic time warping could be used to make time axis better aligned, consider [1] Why international news and china news have different size? Is it true that database itself is of different sizes, or the author did some clipping of the data? If possible, more dataset could be used for performance comparison.[1] Cai, X., et.al. "DTWNet: a Dynamic Time Warping Network", NeurIPS 2019.
Author Response
Point 1: For evaluation, some other metric such as dynamic time warping could be used to make time axis better aligned, consider [1]
[1] Cai, X., et.al. "DTWNet: a Dynamic Time Warping Network", NeurIPS 2019.
Response 1: We analyzed dynamic time warping (DTW) algorithm and found that the DTW algorithm may not be applicable in our task. Firstly, the output of our model is ID of words. The value of ID is meaningless. Words with similar IDs do not have similar meanings. Secondly, DTW algorithm requires the correspondence between points to be monotonic, but the correspondence between word sequences does not have this characteristic. Therefore, DTW may not be suitable for the calculation of word sequences similarity. In section 4.4, for the event selection task, we used Jaccard index to calculate the similarity between the prediction result and candidate events, which aligns the events to some extent. And in event selection task, our model also achieved good results, which also verified the accuracy of the model. Therefore, we calculated the perplexity of different models.
For language models, perplexity can verify the quality of the language model to some degree. However, the probability model and the deep learning model have different methods for calculating the perplexity, and for deep neural networks, we have not found a more standard method for calculating the perplexity. Therefore, to ensure the preciseness, we did not put results into our paper. The perplexity of different models is shown in the table below.
|
|
Perplexity |
|
|
|
International News |
China News |
|
Backoff N-Gram |
583.12 |
811.78 |
|
Modified Kneser-Ney |
275.02 |
378.23 |
|
GRU |
88.23 |
111.05 |
|
HRED |
85.62 |
109.95 |
|
HS-GRU |
72.96 |
105.64 |
Results also show that our model outperforms other models. Adding prior knowledge can help with event prediction.
Point 2: Why international news and china news have different size? Is it true that database itself is of different sizes, or the author did some clipping of the data? If possible, more dataset could be used for performance comparison.
Response 2: Different size between international news and China news is mainly due to the dataset itself. Because Sina News is a mainstream news site in China, it pays more attention to domestic news. For international news, the media usually only focuses on major international events, so the number of international news is less than the number of China news. In line 261 to line 264 of the PDF file, we explained this problem. At the same time, due to the specific task of our model, no benchmark dataset can be applied to the model. For the method of manual constructed, the results often have errors, and large amounts of training data need to be costly. So, no other suitable dataset has been found.

Reviewer 3 Report
Major Comments
In the manuscript, you kept mentioning "time series and prior knowledge", but I do not see how the time series and prior knowledge have been utilized in your analysis. Please show them in your results, taking one or two examples. You could provide how great your lower Error Rates than other existing algorithms. Your method has a broad range of effects to be applied into multiple applications (e.g., traffic jams, disease spreads and so forth). Please have broader conclusions in the Conclusion section.
Minor Comments
Please check your academic English. (e.g., Lu et al., [13] propose -> proposes (line 78). Isn't Figure 1 supposed to be Table 1?
Author Response
Point 1: In the manuscript, you kept mentioning "time series and prior knowledge", but I do not see how the time series and prior knowledge have been utilized in your analysis. Please show them in your results, taking one or two examples.
Response 1: Our event prediction task essentially predicts the sequence of events. So in our model, time series provides event word order information for prediction. The prior knowledge is a supplement to the information obtained by the RNN, which supplements the current prediction by historical events. Therefore, the prediction result is more accurate after adding prior knowledge to the time-based information. We added analysis of experimental results in lines 316, line 318 and line 332. The prediction result is shown in Figure 4, and the result analysis is added in line 340.
Point 2: You could provide how great your lower Error Rates than other existing algorithms. Your method has a broad range of effects to be applied into multiple applications (e.g., traffic jams, disease spreads and so forth). Please have broader conclusions in the Conclusion section.
Response 2: We added broader conclusions in line 374 and line 381 in pdf file. We summarized the mobility of our model in different applications.
Point 3: Please check your academic English. (e.g., Lu et al., [13] propose -> proposes (line 78).
Response 3: We have modified and improved the grammar problem in the paper. For the problem you pointed out, we want to express that many authors, including Lu, propose this method, so we used “propose”.
Point 4: Isn't Figure 1 supposed to be Table 1?
Response 4: Figure 1 wants to show the evolution path of events between different news reports. The display form is similar to the table. At the same time, it does not show the time order between the events. I have modified the figure, as shown in Figure 1 on page 2 of the pdf file.
